# Cost-effectiveness of cell salvage and donor blood transfusion during caesarean section: results from a randomised controlled trial

Carol McLoughlin,[1] Tracy E Roberts,[1] Louise J Jackson,[1] Philip Moore,[2] Matthew Wilson,[3] Richard Hooper,[4] Shubha Allard,[5] Ian Wrench,[6] Lee Beresford,[4] James Geoghegan,[2] Jane Daniels,[7] Sue Catling,[8] Vicki A Clark,[9] Paul Ayuk,[10] Stephen Robson,[11] Fang Gao-Smith,[12] Matthew Hogg,[13] Doris Lanz,[14] Julie Dodds,[14] Khalid S Khan,[14] on behalf of the SALVO study group

**Correspondence to**
Dr Tracy E Roberts;
t.e.roberts@bham.ac.uk

## ABSTRACT

**Objectives** To evaluate the cost-effectiveness of routine use of cell salvage during caesarean section in mothers at risk of haemorrhage compared with current standard of care.

**Design** Model-based cost-effectiveness evaluation alongside a multicentre randomised controlled trial. Three main analyses were carried out on the trial data: (1) based on the intention-to-treat principle; (2) based on the per-protocol principle; (3) only participants who underwent an emergency caesarean section.

**Setting** 26 obstetric units in the UK.

**Participants** 3028 women at risk of haemorrhage recruited between June 2013 and April 2016.

**Interventions** Cell salvage (intervention) versus routine care without salvage (control).

**Primary outcome measures** Cost-effectiveness based on incremental cost per donor blood transfusion avoided.

**Results** In the intention-to-treat analysis, the mean difference in total costs between cell salvage and standard care was £83. The estimated incremental cost-effectiveness ratio (ICER) was £8110 per donor blood transfusion avoided. For the per-protocol analysis, the mean difference in total costs was £92 and the ICER was £8252. In the emergency caesarean section analysis, the mean difference in total costs was £55 and the ICER was £13 713 per donor blood transfusion avoided. This ICER is driven by the increased probability that these patients would require a higher level of postoperative care and additional surgeries. The results of these analyses were shown to be robust for the majority of deterministic sensitivity analyses.

**Conclusions** The results of the economic evaluation suggest that while routine cell salvage is a marginally more effective strategy than standard care in avoiding a donor blood transfusion, there is uncertainty in relation to whether it is a less or more costly strategy. The lack of long-term data on the health and quality of life of patients in both arms of the trial means that further research is needed to fully understand the cost implications of both strategies.

**Trial registration number** ISRCTN66118656.

## Strengths and limitations of this study

► Study strengths of this model-based economic evaluation include that it was based on a rigorously conducted randomised controlled trial (RCT) and it benefited from significant clinical and statistical input throughout its design and development.

► The analyses were conducted from a healthcare perspective, and the cost and outcome data measures incorporated into the model were collected prospectively during the RCT using forms filled out during the preoperative, intraoperative and postoperative phases and at the time of discharge from hospital.

► A limitation of the study is that not all potential outcomes have been included because of the limited time scale in the model and the lack of long-term data.

► A further limitation of the evaluation is that outcomes were expressed in terms of clinical effectiveness rather than in terms of a standard unit of benefit such as the quality-adjusted life year.

## INTRODUCTION

Excessive blood loss (haemorrhage) in childbirth is a life-threatening condition which is an important cause of maternal death,[1] emergency hysterectomy[2] and maternal critical care admission[3] among women undergoing a caesarean section.[4] The treatment for major haemorrhage includes donor blood transfusion. However, red cell concentrates used in donor transfusion are a scarce, nationally pooled resource in demand simultaneously by many clinical services.[5] Furthermore, such transfusions carry risks for recipients as a result of incompatibility or infection.[6] Consequently, there is a recognised need to make caesarean safer while at the same time

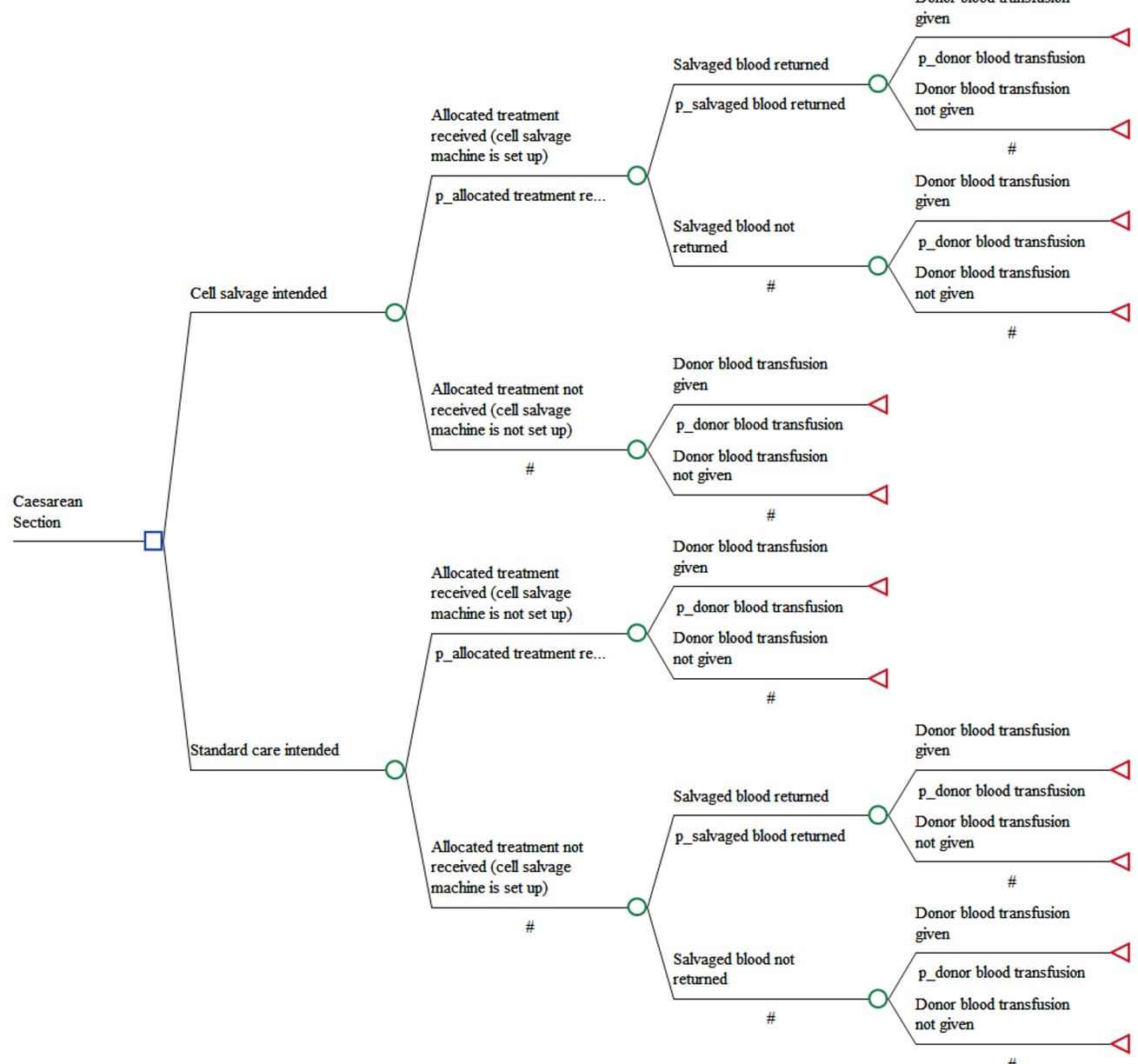

**Figure 1** Decision tree structure.

promoting the use of transfusion alternatives harnessing the patient's own reserves where feasible.[7 8]

Intraoperative cell salvage (IOCS), which collects, processes and returns the woman's own blood lost during surgery, is increasingly being deployed during caesareans. It has been shown to reduce the need for donor blood transfusions in a wide spectrum of surgical disciplines.[9 10]

This study presents the results of a model-based economic evaluation which compared the relative cost and cost-effectiveness of routine cell salvage with standard transfusion with donor blood in caesarean section. This study is part a wider NIHR-HTA randomised controlled trial (RCT) exploring the effectiveness and cost-effectiveness of the use of cell salvage compared with donor blood transfusion during caesarean section.[11] Primary data on costs and resource use were collected prospectively

alongside the trial and the principal outcome for the economic evaluation was the cost per donor blood transfusion avoided.

## METHODS
Full details of the RCT are reported elsewhere.[11] Briefly, the study was performed in 26 obstetric units. The sample comprised women who were admitted to a participating labour ward and who fulfilled all of the following inclusion criteria: aged ≥16 years; able to provide informed consent; undergoing delivery by caesarean section with an identifiable increased risk of haemorrhage, defined as all emergency caesarean sections, where maternal or fetal compromise is suspected, and elective caesarean section for all indications other than maternal request or breech

**Table 1** Probabilities used in the model

| | Trial data | Probability | Distribution |
|---|---|---|---|
| **Cell salvage intended** | | | |
| Cell salvage intended → allocated treatment received (machine is on) | 1432/1498 | 0.96 | Beta |
| Allocated treatment received → salvaged blood returned | 726/1432 | 0.51 | Beta |
| Allocated treatment received → salvaged blood not returned | 703/1432 | 0.49 | Beta |
| Salvaged blood returned → donor blood transfusion given | 22/726 | 0.03 | Beta |
| Salvaged blood returned → donor blood transfusion not given | 704/726 | 0.97 | Beta |
| Salvaged blood not returned → donor blood transfusion given | 9/703 | 0.01 | Beta |
| Salvaged blood not returned → donor blood transfusion not given | 697/703 | 0.99 | Beta |
| Cell salvage intended → allocated treatment not received (machine is off) | 66/1498 | 0.04 | Beta |
| Allocated treatment not received → donor blood transfusion given | 6/66 | 0.09 | Beta |
| Allocated treatment not received → donor blood transfusion not given | 60/66 | 0.91 | Beta |
| Standard care intended | | | |
| Standard care intended → allocated treatment received (machine is off) | 1434/1492 | 0.96 | Beta |
| Allocated treatment received → donor blood transfusion given | 47/1434 | 0.03 | Beta |
| Allocated treatment received → donor blood transfusion not given | 1387/1434 | 0.97 | Beta |
| Standard care intended → allocated treatment not received (machine is on) | 58/1492 | 0.04 | Beta |
| Allocated treatment not received → salvaged blood returned | 35/58 | 0.60 | Beta |
| Allocated treatment not received → salvaged blood not returned | 23/58 | 0.40 | Beta |
| Salvaged blood returned → donor blood transfusion given | 4/35 | 0.11 | Beta |
| Salvaged blood returned → donor blood transfusion not given | 31/35 | 0.89 | Beta |
| Salvaged blood not returned → donor blood transfusion given | 1/23 | 0.04 | Beta |
| Salvaged blood not returned → donor blood transfusion not given | 22/23 | 0.96 | Beta |

presentation. Participating women were randomised to either caesarean section with cell salvage (intervention group) or to caesarean section without cell salvage (control group) for which transfusion of donor blood was carried out according to local guidelines. The primary outcome for the RCT was the proportion of women receiving donor blood transfusion due to haemorrhage. For the economic analysis, a decision analytic model was deemed the most suitable method of presenting the alternative pathways and collating the data for analysis. The economic evaluation took the form of a cost-effectiveness analysis from the perspective of the healthcare provider based on the principal clinical outcome of the trial, that is, the proportion of women receiving donor blood transfusion due to haemorrhage. Given the objectives of the trial and the duration of follow-up, outcomes beyond the trial endpoint were not considered plausible or relevant in the model-based analysis.

### Model structure

A decision tree model was developed in TreeAge Pro 2016 (TreeAge Software, Williamstown, Massachusetts, USA) to represent the alternative strategies. The pathways of the model (see figure 1) represent, as far as possible, the clinical procedures carried out in the study.

### Clinical data used in the model

The model combines the probability of a woman following a particular path and the associated costs. Probabilities, detailed in table 1, were obtained from the trial and attached to each pathway.

### Resource use and costs

The resource use for both arms of the trial was estimated by evaluating the individual components of these procedures (bottom-up costing). Unit cost data were then attached to the resource use. Data were collected on all major National Health Service (NHS) resource use for each patient using the trial case report forms.

For the analysis, intraoperative and postoperative resource use data were obtained from the SALVage in Obstetrics (SALVO) trial. The main resource use monitored included: equipment and disposables required for the cell salvage procedure; additional staff called into theatre solely for the purposes of cell salvage; drugs used in the caesarean section procedure; length and type of hospital inpatient stay including any additional treatment required attributed to the caesarean procedure; the use of donor blood transfusion and salvaged blood transfusion to deal with haemorrhage and its consequences.

Intraoperative costs were estimated for each item to arrive at a mean cost per caesarean procedure for each

treatment pathway in the model. To estimate the cost of a caesarean section procedure, some costs were calculated at the patient level, for example, swab washing, and some at the procedural level, for example, drugs used in the caesarean section procedure (see table 2). Postoperative costs were estimated for each item based on their occurrence in each brand of the model to arrive at a mean cost per patient for each branch (see table 3).

## Analysis

Given the objectives of the trial and the duration of follow-up, a within-trial economic analysis was carried out. The analysis took the perspective of the NHS and Personal Social Services (PSS) following current recommendations from National Institute of Health and Care Excellence (NICE).[12] The main economic analysis was a cost-effectiveness analysis (CEA) with results expressed as cost per donor transfusion avoided.

We carried out three main analyses on the trial data. The first analysis was based on the intention-to-treat (ITT) principle (Analysis ITT). In this method, patients are compared within the treatment groups to which they were originally randomised irrespective of the treatment received.[13] This method of analysis allows the estimates to follow real-life scenarios in which patients may not always receive the planned treatment. ITT is recommended to ensure that the benefits of a given intervention are not exaggerated.[13] The second analysis was based on the treatment actually received by patients irrespective of randomisation (Analysis 'per protocol' (PP)). Within the SALVO trial, equal numbers of patients were randomised to either cell salvage or standard care. However, because some clinicians managing women in the control arm had access to a cell salvage machine, it was possible that women in the control arm could receive cell salvage in place of a donor blood transfusion. A PP analysis was carried out to look at the effect of treatment received on the outcome estimates. Therefore, in Analysis PP, all patients who received cell salvage were compared with those who received standard care, irrespective of the treatment to which they were randomised. The third analysis considered only patients who underwent an emergency caesarean section (Analysis ECS). This analysis was considered necessary as the SALVO trial found that numerically, there was a greater reduction in the rate of transfusion within the emergency patient group compared with the elective patient group. This analysis followed the same methodology as the ITT and PP analyses. Deterministic and probabilistic sensitivity analyses (PSAs) were carried out for each analysis to explore the effects of the inherent uncertainty in parameter estimates on model results.

## Patient and public involvement

The UK National Childbirth Trust collaborated in the project by providing patient and public input through involvement in trial design and protocol development. Prior to the SALVO trial, a survey was conducted among women who received cell salvage, showing that they perceived the interventions as reassuring, safe and preferable to donor blood transfusion. A patient representative was a member of the trial steering committee to provide oversight and advice regarding recruitment, dissemination and general trial management. We are planning to disseminate findings to participants in the form of a newsletter following primary publication of these results.

## RESULTS

The results of the ITT analysis (Analysis ITT) suggest that routine cell salvage is more costly than standard care with the mean difference in total costs per patient estimated at £83. The incremental cost-effectiveness ratio (ICER) representing the additional cost of routine cell salvage during caesarean section in women at risk of haemorrhage compared with standard care for these women was estimated to be approximately £8110 per donor blood transfusion avoided. The overall result, based on the PP analysis (Analysis PP), was an ICER of £8252 per transfusion avoided for cell salvage compared with standard care. The ECS (Analysis ECS) resulted in an ICER of £13 713 per transfusion avoided. See table 4.

The results of the corresponding PSAs are presented using cost-effectiveness acceptability curves (CEACs) to graphically represent uncertainty in the appropriate threshold cost-effectiveness value (figure 2). The CEAC for Analysis ITT shows that the probability that routine cell salvage is cost-effective increases as the willingness to pay for a donor blood transfusion avoided increases. If the maximum willingness to pay for a donor blood transfusion avoided was £50 000, for example, the probability of routine cell salvage being cost-effective would be 62%. For Analysis PP, the probability that routine cell salvage was cost-effective at a willingness to pay £50 000 would be 63%. The CEAC for Analysis ECS shows that the probability that cell salvage is cost-effective remains between 47% and 55% as the willingness to pay for a donor blood transfusion avoided increases.

## Deterministic sensitivity analysis

A number of deterministic sensitivity analyses were conducted in each analysis (see table 5):

i. The main analyses used costs for consumables based on a particular model of the cell saver machine. To assess the difference that variation in these estimates would make, the unit costs were replaced with unit costs obtained from the NICE costing statement for blood transfusion.[14] This had a marginal impact on the ICER in each analysis. The impact of the inclusion of acquisition costs for a cell salvage machine was also explored and found this had only a minimal impact on the ICER in each analysis. In the trial, 202 cases used a cell salvage machine that required consumables for collection only, even where the blood was not processed. The impact of including costs for the consumables used by this machine, where the machine is only set up for processing in patients

**Table 2** Intraoperative resource use and costs

| Item | Resource use Cell salvage (n=1498) | Control (n=1492) | Unit cost | Mean cost per procedure Cell salvage (n=1498) | Control (n=1492) | Assumption/Working | Source |
|---|---|---|---|---|---|---|---|
| Running costs | 1432 | 58 | £6.14 | £6.14 | £6.14 | Based on annual maintenance costs for Haemonetics Cell Saver 5 (Haemonetics UK) machine and estimated annual usage | UHB, (Mr Scott Hancock, University Hospitals Birmingham, 2016, personal communication); NICE costing statement blood transfusion (November 2015)[14] |
| Collection set | 1 | 1 | £41.71 | £41.71 | £41.71 | Based on the assumption that one collection set is used per procedure | NHS Supply Chain Catalogue Autotransfusion reservoir 3 L[24] |
| Processing pack | 1 | 1 | £77 | £77 | £77 | Based on the assumption that one processing pack is used per procedure | NHS Supply Chain Catalogue Intraoperative autologous blood system cell saver 5+ bowl set 125 mL (Haemonetics UK)[24] |
| Leucocyte depletion filter | 782 | 25 | n/a | n/a | n/a | Cost not included in the analysis as leucocyte depletion filter included in the collection set for Haemonetics Cell Saver 5 machine (Haemonetics UK) | NHS Supply Chain Catalogue Autotransfusion reservoir 3 L[24] |
| Additional sucker | 598 | 29 | £15.41 | £6.43 | £7.70 | Mean cost based on the number of additional suckers used in each treatment arm/total number of patients who received cell salvage | NHS Supply Chain Catalogue Aspiration and anticoagulation line Cell Saver (Haemonetics UK) £308.02 for 20[24] |
| Swab washing | 781 | 21 | £0.80 | £0.44 | £0.29 | Mean cost based on the number of times swabs were washed in each treatment group/total number of patients who received cell salvage | ICS Factsheet 1 Swab Washing March 2015,[25] based on the cost of 1L of sodium chloride 0.9%, BNF[26] |
| Staff | | | £0.72 (min) | £11.57 | £12.03 | Based on the staff type most frequently called into theatre solely for the purposes of cell salvage | Unit cost for hospital based nurse, band 5, PSSRU unit costs 2015 (costs include qualifications)[27] |
| Saline (L) | 2 | 2 | £0.80 | £1.60 | £1.60 | Based on the assumption that 2 L of saline would be administered to all patients undergoing cell salvage prior to collection[14] | Based on the cost of 1 L of 0.9% sodium chloride, BNF[26] |
| Heparin sodium (30 000 IU) | 2 | 2 | £10.60 | £21.20 | £21.20 | Based on the assumption that 60 000 IU of heparin would be administered to all patients undergoing cell salvage prior to collection[14] | Based on the cost of 1 mL amp of heparin sodium 25 000 IU/mL and 1 mL amp of heparin sodium 5000 IU/mL, BNF[26] |

Continued

**Table 2** Continued

| Item | Resource use | | Unit cost | Mean cost per procedure | | Assumption/Working | Source |
|---|---|---|---|---|---|---|---|
| | Cell salvage (n=1498) | Control (n=1492) | | Cell salvage (n=1498) | Control (n=1492) | | |
| Anti-D (500 IU) | 1 | 1 | £33.75 | £3.04 | £3.04 | Based on the assumption that all RhD-negative women delivering a RhD-positive baby receive at least 500 IU of anti-D.[28] Mean cost per procedure based on the probability of a woman requiring anti-D in each treatment arm (0.09) | Based on the cost of 500-unit phial of anti-D immunoglobulin, BNF[26] |
| Anti-D (1500 IU) | 1 | 1 | £58 | £5.22 | £5.22 | Based on the assumption that women who receive cell salvage are offered 1500 IU of anti-D.[28] Mean cost per procedure based on the probability of a woman requiring anti-D in each treatment arm (0.09) | Based on the cost of 1500-unit phial of anti-D immunoglobulin, BNF[26] |
| RBC transfusion (units) | 3 | 3 | First unit: £194 Subsequent units: £166 | £520 | £520 | Based on the assumption that all units transfused in each treatment arm were RBC[14] | NICE costing statement for blood transfusion (November 2015).[14] Unit cost for RBCs obtained from NHSBT 2016/17[15] |

BNF, British National Formulary; NHSBT, National Health Service Blood Transfusion; NICE, National Institute for Clinical and Health Excellence; PSSRU, Personal Social Services Research Unit; RBC, Red Blood Count.

**Table 3** Postoperative resource use and costs

| Item | Resource use | | | Mean cost per patient | | | Source |
|---|---|---|---|---|---|---|---|
| | Cell salvage (n=1498) | Control (n=1492) | Unit cost | Cell salvage (n=1498) | Control (n=1492) | | |
| Inpatient stay (normal days) | 3734.5 | 3852 | £431.45 | £1074 | £1113 | | NHS reference costs 2014/15.[29] Weighted average unit cost for elective and non-elective inpatient bed days |
| Inpatient stay (HLC) | 189.5 | 136 | £539–£848* | £78 | £56 | | NHS reference costs 2014/15[29] National tariff payment system 2016/17[30] |
| Adverse events | 3 | 0 | n/a | n/a | n/a | | Assumption that transfusion would be discontinued in the event of an adverse reaction based on BCSH guidelines[31] |
| Hospital transfer | 2 | 2 | £99 | £0.13 | £0.13 | | PSSRU 2015[27] |
| Investigations | 6 | 10 | £94–£138† | £0.42 | £0.70 | | NHS reference costs 2014/15[29] |
| Additional surgery | 11 | 8 | £399–£2991‡ | £13 | £9 | | NHS reference costs 2014/15[29] |
| RBC transfusion (units) | 3 | 3 | First unit: £194 Subsequent units: £166 | £13 | £17 | | NICE costing statement for blood transfusion (November 2015). Unit cost for RBC obtained from NHSBT 2016/17[15] |
| Total cost of postnatal care per patient | | | | £1178.55 | £1195.86 | | |

*Range based on cost per day of care: level 1 £539, level 2 £674, level 3 £848.
†Range based on unit cost of a CT scan (£94) and an MRI scan (£138).
‡Range based on unit cost of additional surgeries (less cost of days in hospital).
BCSH, Blood Components. British Committee for Standards in Haematology; NHSBT, National Health Service Blood Transfusion; NHS, National Health Service; NICE, National Institute for Clinical and Health Excellence; PSSRU, Personal Social Services Research Unit; RBC, Red Blood Count.

**Table 4**  Results

| Transfusion strategy | Average cost per patient (£) | Difference in costs (£) | Effectiveness (donor blood transfusion avoided) | Incremental donor blood transfusion avoided | ICER per donor blood transfusion avoided (£) |
|---|---|---|---|---|---|
| Analysis ITT | | | | | |
| Standard care | 1244 | | 0.965 | | |
| Cell salvage | 1327 | 83 | 0.975 | 0.010 | 8110 |
| Analysis PP | | | | | |
| Standard care | 1238 | | 0.967 | | |
| Cell salvage | 1330 | 92 | 0.978 | 0.011 | 8252 |
| Analysis ECS | | | | | |
| Standard care | 1352 | | 0.986 | | |
| Cell salvage | 1407 | 55 | 0.990 | 0.004 | 13 713 |

ECS, emergency caesarean section; ICER, incremental cost-effectiveness ratio; ITT, intention to treat; PP, per protocol.

having blood returned and where swab washing is not conducted, resulted in an ICER of £1022 in Analysis ITT, £1184 in Analysis PP. In Analysis ECS, there was a dominant result in that cell salvage was considered less costly and more effective compared with standard care.

ii. The base case analyses used the mean length of time additional staff were present in theatre in each arm solely for the purposes of cell salvage. In the sensitivity analysis, the cost of additional staff was removed. This reduced the ICER to £7065 in Analysis ITT, £7210 in Analysis PP and £10 932 in Analysis ECS.

iii. To facilitate robust evaluation in cost-effectiveness analyses relating to donor blood, a comprehensive estimate for the cost of a unit of donor blood is required. The NHS Blood and Transplant Authority have valued the cost of RBC to be £120 per unit based on direct costs to the healthcare services.[15] However, there is significant uncertainty surrounding this

figure. We conducted a study (submitted for publication) parallel to the SALVO trial that aimed to dissect the current price of blood. We explored what elements are contributing to the current cost of blood and what elements are missing. Our study concluded that the current costing approach of assuming there will always be an adequate supply of donor blood must be replaced with including provisions for the continued shrinking of the donor pool and the impact that future shocks to the blood supply system could have. This sensitivity analysis assessed the difference that variation in the estimated cost of blood made to the overall cost-effectiveness of routine cell salvage. Raising the cost of a three-unit transfusion of RBC to a hypothetical cost of £1500 reduced the ICER by £974 in both the ITT and PP analysis and it reduced the ICER by over £1000 in the ECS. Threshold analysis showed that for routine cell salvage to be considered cost-effective, the cost of an RBC transfusion would have to increase to £8637 in Analysis ITT, £8778 in Analysis PP and £13 186 in Analysis ECS.

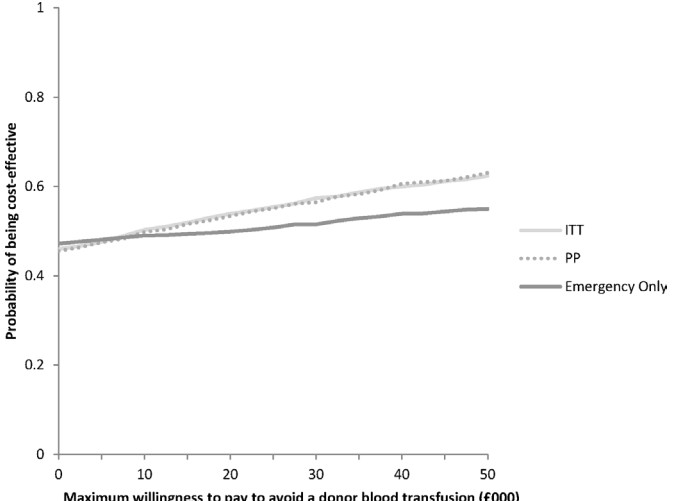

**Figure 2**  Cost-effectiveness acceptability curve for donor blood transfusion avoided. ITT, intention to treat; PP, per protocol.

## DISCUSSION
### Principal findings
The results of this economic evaluation suggest that routine cell salvage is more costly than standard care with the average cost per patient estimated at £1327. The ICER for this strategy compared with standard care is approximately £8110 per donor blood transfusion avoided. The PSA suggests that at an arbitrary willingness to pay threshold of £50 000, the probability of routine cell salvage being cost-effective is 62%.

The results of this analysis were shown to be robust for the majority of deterministic sensitivity analyses with one exception; using a cell salvage machine that required different consumables to those included in the main base case analyses, where the machine is only set up for processing in patients having blood returned and where

**Table 5** Deterministic sensitivity analysis

| | Original value | Revised value | Cost per donor blood transfusion avoided | | |
| --- | --- | --- | --- | --- | --- |
| | | | **Original result** | | |
| | | | Analysis intention to treat **£8110** | Analysis per protocol **£8252** | Analysis emergency caesarean section **£13713** |
| | | | **Revised result** | | |
| **(1) Equipment and disposables required for the cell salvage procedure** | | | | | |
| Applying the cost of consumables (collection set+processing pack) used by National Institute of Health and Care Excellence | £118.71 | £119.75 | £8205 | £8346 | £13952 |
| Including acquisition costs based on Haemonetics Cell Saver machine | – | £22.13 | £10114 | £10246 | £18805 |
| Using a continuous transfusion cell saver where the machine is only set up for processing in patients having blood returned and swab washing is not conducted | Cell salvage £125.14 Control £126.40 | Cell salvage £34.73 | £1022 | £1184 | Dominates |
| **(2) Staff** | | | | | |
| No additional member of staff being called into theatre solely for the purposes of cell salvage | Cell salvage £11.57 Control £12.03 | Cell salvage £0 Control £12.03 | £7065 | £7210 | £10932 |
| **(3) Donor blood** | | | | | |
| Variation in the estimate of the cost of a three-unit RBC transfusion | £520 | £750 £1000 £1250 £1500 | £7886 £7636 £7386 £7136 | £8028 £7778 £7528 £7278 | £13330 £13080 £12830 £12580 |
| For routine cell salvage to be considered cost-effective in this model, the price of a three-unit RBC transfusion would have to be: | | | £8637 | £8778 | £13186 |

swab washing is not conducted, resulted in a significant effect on the ICER, reducing it to £1022 per donor blood transfusion avoided.

A PP analysis produced an ICER of £8252 per transfusion avoided, but this result should be considered with caution as the population in this analysis is a subset of the ITT population who completed the study without any major protocol violations.[13] In clinical practice, uptake of cell salvage is unlikely to be 'per-protocol'. In a third analysis, looking at emergency caesareans only, cell salvage appears to be more effective than standard care for avoiding a donor blood transfusion but the resulting ICER of £13 713 is driven by the increased probability that these patients will require a higher level of postoperative care and additional surgeries.

### Strengths and limitations of the study

The strength of this model-based economic evaluation is that it was based on a rigorously conducted RCT. The cost and outcome data measures that were incorporated into the model were collected prospectively during the RCT using forms filled out at the preoperative, intraoperative and postoperative phase and at the time of discharge from hospital. In addition, the economic evaluation benefited from significant clinical and statistical input throughout its design and development. All assumptions used in the model were agreed with the trial team before the analysis was carried out and without knowledge of how these assumptions would affect the results.

In terms of limitations, it was not possible to account for long-term implications relating to maternal fetal exposure as data relating to this were not available from the trial. In addition, information relating to the clinical status and care of the infant was not included in the analysis. A further limitation of the study is that outcomes were expressed in terms of clinical effectiveness rather than in terms of a standard unit of benefit such as the quality-adjusted life year. Finally, the use of platelets and other blood products has not been included in the study. However, the results of the sensitivity and threshold analyses demonstrated that including these costs would not have impacted on the cost-effectiveness results.

### Strengths and weaknesses in relation to other studies

To date, there has only been one, small RCT looking at the elective use of cell salvage at caesarean section[16] and this study did not include an economic element. A Cochrane review of cell salvage in adult elective surgery assessed the clinical and cost-effectiveness of cell salvage and other autologous transfusion strategies in elective surgery.[10] It suggested that cell salvage may be an 'effective and cost-effective alternative to the allogeneic blood transfusion strategy'. However, no obstetric papers were identified for this review. A recent study by Lim *et al*[17] found the use of cell salvage for cases at high risk for obstetric haemorrhage to be economically reasonable while routine cell salvage use for all caesarean deliveries was not. In contrast to the current study, Lim *et al* adopted

a societal perspective and used data from published research to populate the model.[17]

### Meaning of the study

The results of the economic evaluation suggest that while routine cell salvage is a marginally more effective strategy than standard care in avoiding a donor blood transfusion, there is uncertainty in relation to whether it is a less or more costly strategy. Under the conditions reported here, for a high-income country such as the UK, where donor blood is typically available, cell salvage is unlikely to be considered a cost-effective alternative to the provision of donor blood by the health service. However, in lower/middle-income countries where the provision of a safe and secure blood supply may be more challenging, the relative cost-effectiveness may be very different. In addition, the lack of long-term data on the health and quality of life of patients in both arms of the trial means that further research is needed to fully understand the cost implications of both strategies. For example, latent infection such as hepatitis may result in chronic liver disease within 20 or more years of incident of infection[18 19] which has obvious long-term cost implications for the healthcare provider.

### Unanswered questions and future research

The current study has used data from a large, multicentre randomised trial which demonstrated modest evidence that routine use of cell salvage during caesarean section reduced the need for donor blood transfusion. The main cause of uncertainty relates to the long-term cost implications of adopting the routine use of cell salvage. Future studies should explore the long-term health and economic and quality of life impacts associated with both transfusion strategies. Also, evidence on the preferences of women needs to be considered. For example, hospitals may wish to have the option of cell salvage available for Jehovah witness patients where there is no option to use donor blood. In countries where safe donor supply cannot be guaranteed the use of cell salvage might have very different implications which need to be explored.

Finally, the issue of donor blood as a scarce resource needs to be considered. As things currently stand, demand for donor blood is increasing, while on the other hand, enhanced safety measures are limiting the donor pool.[20 21] The impact of further restrictions on supply could create shortages under current usage patterns, and donor blood substitutes such as cell salvage play a potential role in helping to re-establish a demand–supply balance.[22 23] Not considered in this study is the fact that transfusion with cell salvage can always exist. While there is an expectation that donor blood will always be there when needed, transfusion using donor blood simply cannot be guaranteed. In such a scenario, where the option of donor blood is limited or not available, the routine use of cell salvage would be dominant (less costly and more effective) compared with standard care, thus making provision

for the availability of the technology likely to be extremely important.

**Author affiliations**
[1]Health Economics Unit, University of Birmingham, Birmingham, UK
[2]Selwyn Crawford Department of Anaesthetics, Birmingham Women's Hospital, Birmingham, UK
[3]School of Health and Related Research (ScHARR), University of Sheffield, Sheffield, UK
[4]Pragmatic Clinical Trials Unit, Centre for Primary Care & Public Health, Queen Mary University of London, London, UK
[5]Haematology, Barts Health NHS Trust and NHS Blood and Transplant, London, UK
[6]Departmentof Anaesthesia (Northern General Hospital), SheffieldTeaching Hospitals NHS Foundation Trust, Sheffield, UK
[7]Nottingham Clinical Trials Unit, University of Nottingham, Nottingham, UK
[8]Department of Obstetrics, Singleton Hospital, Swansea, UK
[9]Simpson Centre for Reproductive Health, Royal Infirmary of Edinburgh, Edinburgh, UK
[10]Women's Services, RoyalVictoria Infirmary, Newcastle upon Tyne, UK
[11]Institute of Cellular Medicine, Newcastle University, Newcastle Upon Tyne, UK
[12]Perioperative, Critical Care and Trauma Trials Group, University of Birmingham, Birmingham, UK
[13]Barts Health NHS Trust, Royal London Hospital, London, UK
[14]Women's Health Research Unit, Barts and The London School of Medicine and Dentistry, Queen Mary University of London, London, UK

**Acknowledgements** The study group would like to thank local midwifery, theatre obstetric and anaesthetic teams for their support, as well as members of the trial steering and data monitoring committees, patient and public representatives, staff at the Women's Health Research Unit and Pragmatic Clinical Trials Unit at Queen Mary University of London and all the women who agreed to participate in the SALVO trial.

**Contributors** CM undertook the economic analysis, interpreted the results and drafted the manuscript with input from all authors. TER contributed to the trial design, developed the economic analysis plan, supervised the economic analysis and revised the paper. LJJ contributed to the economic analysis plan and revised the paper. PM contributed to the trial design, provided clinical advice and revised the paper. MW contributed to the trial design and revised the paper. RH contributed to the trial design, developed the statistical plan and revised the paper. SA contributed to the trial design and revised the paper. IW contributed to the trial design and revised the paper. LB performed the trial statistical analysis and revised the paper. JG, JD, SC, VAC, PA, SR, FG and MH contributed to the trial design and revised the paper. DL supervised the general management of the trial, supervised data collection and revised the paper. JD supervised the running of the trial and revised the paper. KK was the chief investigator for the trial, contributed to the trial design, managed the trial and revised the paper.

**Funding** Funding for this study was provided by the Health Technology Assessment Programme of the National Institute for Health Research. Project number 10/57/32.

**Disclaimer** The views and opinions expressed herein are those of the authors and do not necessarily reflect those of the Health Technology Assessment programme, NIHR, NHS or the Department of Health.

**Competing interests** None declared.

**Patient consent** Obtained.

**Provenance and peer review** Not commissioned; externally peer reviewed.

**Data sharing statement** No additional data available.

**Author note** Consolidated Health Economic Evaluation Reporting Standards (CHEERS) statement provided.

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
