## [Reviewer comments · BMJ Open]

This paper was submitted to a another journal from BMJ but declined for publication following peer review. The authors addressed the reviewers' comments and submitted the revised paper to BMJ Open. The paper was subsequently accepted for publication at BMJ Open.

(This paper received three reviews from its previous journal but only two reviewers agreed to published their review.)

ARTICLE DETAILS

TITLE (PROVISIONAL)	Cost-effectiveness of cell salvage and donor blood transfusion during caesarean section: results from a randomised controlled trial
AUTHORS	McLoughlin, Carol; Roberts, Tracy; Jackson, Louise J; Moore, Philip; Wilson, Matthew; Hooper, Richard; Allard, Shubha; Wrench, Ian; Beresford, Lee; Geoghegan, James; Daniels, Jane; Catling, Sue; Clark, Vicki A; Ayuk, Paul; Robson, Stephen; Gao, Fang; Hogg, Matthew; Lanz, Doris; Dodds, Julie; Khan, Khalid

VERSION 1 – REVIEW

REVIEWER	Grace Lim MD MS University of Pittsburgh, Magee-Womens Hospital, USA
REVIEW RETURNED	29-Mar-2018

GENERAL COMMENTS	Major strengths. The authors describe a cost-effectiveness analysis for the use of cell salvage for patients at high risk for hemorrhage, that was conducted in parallel with a randomized control trial (SALVO). The writing is clear and the methods for the original randomized trial are strong. Major weaknesses. The conclusion that cell salvage is not cost-effective cannot be substantiated by the investigators' data. As identified by the authors, the major weakness is the lack of QOL assessment which precludes the ability to take a societal perspective in the analysis using the QALY utility. The analysis at the healthcare perspective level is outside of the recommendations from the panel on cost-effectiveness in health and medicine (see reference below), and although this approach is acceptable, it limits the ability to make broader statements about the cost-effectiveness of an intervention beyond a healthcare perspective. Some discussion about why the healthcare perspective was chosen over a societal one should be included in the manuscript. The investigators did not use QALYs at all, hence the utility of any threshold value is unclear. It is also not clear what the time horizon of their model is – presumably, it is the length of the trial – but again this limits broader statements about cost-effectiveness.
---

These are major limitations that, unfortunately, markedly limit the usefulness of this new work.

The major limitations described above need to be more thoroughly treated in the manuscript and in the abstract.

The cost-effectiveness acceptability curves are visually very different from what one would expect for this type of curve. Typically, one can easily view the comparison between two or more strategies. Please revisit/check to ensure they are more consistent with traditional output (see Fenwick et al Health Econ. 10: 779 – 787 (2001)).

These results, to a reader who is not familiar with CEA, may seem to conflict with those reported by Lim et al (Anesthesiology, 2018 Feb;128(2):328-337). In the discussion, please discuss how and why the present results may differ from the Lim article (e.g. latter article took societal perspective, QALY outcome), and how they may be similar. In the discussion, for the lay-reader, some explanation for why an approach may look costlier at the healthcare level but overall be cost-effective at the societal level should be included. This study and that of Lim et al. both suggest that the utility of the health state associated with transfusion and autologous blood reinfusion needs to be better defined, with the present study suggesting it by omission.

Page 13 Line 274-276. I am not sure this statement is substantiated by the data, especially because only the healthcare perspective was assumed in this particular study. Please, edit or remove.

Specific comments:

The word “cesarean” is form the Greek “to cut” and thus the word “section” that follows is redundant as it translates to “cut cut.” Consider following all “cesarean” by the word “delivery” instead, throughout the text.

Methods. (Abstract and main text). It is easier for a reader if “Analysis 1, 2, and 3” can be termed something else that ties immediately to their group. E.g. “Analysis IIT” for intent to treat, “Analysis PP” per protocol, and “Analysis ECD” for emergency cesarean delivery. Please, consider.

Methods. Some justification for why the perspective of health care provider, rather than societal perspective, was taken is desirable. Recommendations from the Panel on Cost-Effectiveness in Health and Medicine suggest that a more useful perspective from the standpoint of informing best practice is a societal perspective, rather than a healthcare-level perspective.

Weinstein, MC, Siegel, JE, Gold, MR, Kamlet, MS, Russell, LB . Recommendations of the panel on cost-effectiveness in health and medicine. JAMA 1996; 276:1253–8

“Patient and public involvement” section. It is unclear why this section is here as the content is not germane to the economic analysis provided in the article. Consider removing, please.

REVIEWER	Steven M. Frank MD Johns Hopkins Medical Institutions United States Advisory Boards for two companies that manufacture cell salvage equipment: Medtronic and Haemonetics
REVIEW RETURNED	08-Apr-2018

GENERAL COMMENTS	This randomized clinical trial reports the cost effectiveness of employing cell salvage for caesarean-section patients compared to the routine practice of giving allogeneic banked blood transfusions. Overall the study is large and well conducted, however some improvements in the paper listed below would be helpful to justify and support the main findings - that cell salvage was only marginally more effective in terms of avoiding allogeneic transfusion with uncertain cost effectiveness. 3,028 women were enrolled with a transfusion rate of about 3-5%, depending on the group assignment. Of those assigned to the intervention group (cell salvage), 50.8% of those had salvaged blood returned with a mean volume of 260 mL. The problem is that these numbers are not apparent in the current paper under review but I obtained them from the authors' previous publication of the same study back in December, 2017 (Khan KS, Moore PAS, Wilson MJ, et al. Cell salvage and donor blood transfusion during cesarean section: A pragmatic, multicentre randomised controlled trial (SALVO) PLoS Med 2017;14:e1002471). The paper currently under review is the same exact study only it represents the economic analysis. One concern is that in the original paper on page 12 and in the S1 appendix (pages 2, 4, 6, 19, and 20) the authors already published much of the same economic data presented in the current paper. It would be nice to point out how these findings differ and warrant what could be considered re-publication. Although the findings are included in the previous publication, it would be helpful for the reader to see in the current paper the percentage of patients requiring transfusion of banked blood and the mean number of units transfused per patient in the two groups. I think table 1 may contain the percentage of patients transfused but it is difficult to figure out from the data in the table. I know the current paper is all about cost data, but some blood data also need to be included for clarity. The inclusion criteria should be better described. For example, at LM 120, what were the identifiable risks of haemorrhage and the definition of an emergency caesarean section? In the discussion, the authors may want to reference the following publication on the economic value of cell salvage in comparison to their findings (Waters JH, Dyga RM, Waters JF, Yazer MH. The volume of returned red blood cells in a large blood salvage program: where does it all go? Transfusion 2011;51:2126-32). This paper suggests that if one unit of salvaged blood is reinfused then the salvage procedure is cost effective, and the current study under review suggests this is not the case. Perhaps the difference is the high cost estimate in the current study of £1,244 extra cost per case where cell salvage is employed (stated on LM 204), a number which is not readily apparent from the data shown in Tables 2, 3, and 4. Does this number represent the incremental additional cost of employing cell salvage or just the average cost per case where it is used? The sentence starting on LM 203
---

	doesn't make this clear. Also, does this relatively high cost include having a cell saver technician in the OR the entire case, or just for the initial setup and for processing the blood? This is really important because one cell saver tech can staff multiple ORs as they do not need to be in the OR the entire case. In fact, this variable could make or break the cost effectiveness of cell salvage and should be considered to be important for the current study. This detail should be added to the methods and discussion. On LM 233 the sentence describes 202 centers in the trial but there were only 26 centers so this may be a typo. In this same sentence, a cell salvage machine is described that does not require consumables, and I think this sentence is misleading. All salvage machines require consumables that come in two packages. About 1/3 of the costs go towards the package that always needs to be opened and utilized which is the blood collection reservoir and the heparin solution to anticoagulate the shed blood. The other 2/3 of the costs goes towards the blood processing kit, containing the centrifuge bowl that the cell salvage machine uses for washing the shed blood. This package only need be opened if there is sufficient blood shed during the case to warrant processing. When a case is done under the "collect only" protocol, often only the first package need be opened initially, then in specific higher blood loss cases the second package can be opened. I think is what the authors mean to say when they describe a machine that doesn't require disposables, which should be clarified. This collect only strategy is actually one that can substantially increase the cost effectiveness of using cell salvage and this should be recognized by the authors and perhaps more strongly emphasized. On LM 258 the authors describe variation in the estimated cost of blood and the potential cost increase to £1,500 for 3 units. It is unclear in the text, but what I think what the authors describe here is the full cost of an allogeneic blood unit including the overhead costs of managing that unit from donor to recipient - which is about 4 times higher than the acquisition cost of blood. The authors should reference this paper to support these overhead costs (Shander A, Hofmann A, Ozawa S, et al. Activity-based costs of blood transfusions in surgical patients at four hospitals. Transfusion 2010;50:753-65). On pages 30, 31, and 32, these three figures could all be condensed to one figure using 3 different colored lines to show the relative comparison between the three analyses.
--	--

REVIEWER	Cyril Jacquot Children's National Health System, Washington, DC, United States
REVIEW RETURNED	04-May-2018

GENERAL COMMENTS	This paper evaluates the option and cost-effectiveness of intraoperative cell salvage in obstetric patients as part of patient blood management in the setting of continued shrinking of blood donor pool. Comments: Did all women in the cell salvage arm receive reinfused cells or was there a minimum volume cutoff? For example, in a study by Milne et al. (Obstet Gynecol. 2015 Apr;125(4):919-23), only 21%
---

	of women had sufficient amount of intraoperative blood shed to proceed with reinfusion. In your study of patients with higher risk of bleeding, did the rate reach 100%? Page 7, line 120: Can you list the identifiable risk factors for haemorrhage used? Which ones had particularly good predictive values? Page 10, lines 195-196: In the UK, must blood consenting include discussion of alternatives to allogeneic blood (e.g. autologous blood)? Is that a factor in patients perceiving cells salvage as "reassuring, safe and preferable to donor blood transfusion"? A similar scenario I can think of in the United States is the Paul Gann Blood Safety Act in California: "a physician [...] shall inform [...] the patient of the positive and negative aspects of receiving autologous blood and directed and nondirected homologous blood from volunteers. For purposes of this section, the term "autologous blood" includes, but is not limited to, predonation, intraoperative autologous transfusion, plasmapheresis, and hemodilution." Page 11, lines 233-234 and page 13, lines 279-283: Cell salvage machines that "did not require consumables" and that "required different consumables" are described. I see the same effect on the ICER of £1,022 per donor blood transfusion avoided. Do these sections refer to the same alternative cell salvage machine? If necessary, please clarify the statements about the consumables. Page 15, lines 309-310: I see the platelet/plasma use was not included. I was just curious if you would expect use to differ between patients receiving cell salvage versus donor blood transfusion. Might one group be at risk of more coagulopathy based on the red blood cells transfused?
--	---

VERSION 1 – AUTHOR RESPONSE

Reviewer: 1 Reviewer Name: Grace Lim MD MS Institution and Country: University of Pittsburgh, Magee-Womens Hospital, USA Please state any competing interests: None		
1	The authors describe a cost-effectiveness analysis for the use of cell salvage for patients at high risk for hemorrhage, that was conducted in parallel with a randomized control trial (SALVO). The writing is clear and the methods for the original randomized trial are strong.	Thank you
2	The conclusion that cell salvage is not cost-effective cannot be substantiated by the investigators' data. As identified by the authors, the major weakness is the lack of QOL assessment which precludes the ability	Thank you for your comment. We respectfully disagree that the conclusion cannot be substantiated by the data. Within the context of this trial and based on the

to take a societal perspective in the analysis using the QALY utility. The analysis at the healthcare perspective level is outside of the recommendations from the panel on cost-effectiveness in health and medicine (see reference below), and although this approach is acceptable, it limits the ability to make broader statements about the cost-effectiveness of an intervention beyond a healthcare perspective. Some discussion about why the healthcare perspective was chosen over a societal one should be included in the manuscript. The investigators did not use QALYs at all, hence the utility of any threshold value is unclear. It is also not clear what the time horizon of their model is – presumably, it is the length of the trial – but again this limits broader statements about cost-effectiveness. These are major limitations that, unfortunately, markedly limit the usefulness of this new work. The major limitations described above need to be more thoroughly treated in the manuscript and in the abstract.	results presented it is unlikely that cell salvage would be considered cost-effective. Given the objectives of the trial and the duration of follow up, a within trial economic analysis was carried out as stated on line 170. The analysis was conducted alongside a rigorously conducted randomised controlled trial in the UK. The analysis is conducted from the perspective of the NHS and PSS as recommended by the National Institute for Health Care Excellence. This is stated on line 171 and a reference is provided. The societal perspective here would have required out of pocket costs to the women as well as longer term impact and risk of blood transfusion. Given that the focus of the trial was on the reduction of blood transfusion within the hospital setting, a wider perspective would not have added helpful information to the research problem being addressed. We appreciate the reviewers concern about not using QALYs in this analysis. We have already acknowledged that QALYS were not appropriate for this project as it wasn't within the scope of this study, and identified the limitations of this (line 317). However, based on the within trial analysis, a use of QALYS would not have impacted on the cost-effectiveness results in this study. Given that there is not a pre-specified threshold of willingness to pay for a blood transfusion avoided an arbitrary threshold of £50,000 was chosen in this study. Ultimately we agree with the reviewer that the inclusion of QALYs and a societal perspective is the desired outcome for such analyses. However based on the within trial
---	--

		analysis presented in this study, we believe our results are supportable and the limitations would not have been captured from the within trial analysis
3	The cost-effectiveness acceptability curves are visually very different from what one would expect for this type of curve. Typically, one can easily view the comparison between two or more strategies. Please revisit/check to ensure they are more consistent with traditional output (see Fenwick et al Health Econ. 10: 779 – 787 (2001)).	Thank you for this useful comment. We have now condensed the three figures into one figure using three different colour/textured lines to show the relative comparison between the three analyses.
4	These results, to a reader who is not familiar with CEA, may seem to conflict with those reported by Lim et al (Anesthesiology, 2018 Feb;128(2):328-337). In the discussion, please discuss how and why the present results may differ from the Lim article (e.g. latter article took societal perspective, QALY outcome), and how they may be similar. In the discussion, for the lay-reader, some explanation for why an approach may look costlier at the healthcare level but overall be cost-effective at the societal level should be included. This study and that of Lim et al. both suggest that the utility of the health state associated with transfusion and autologous blood reinfusion needs to be better defined, with the present study suggesting it by omission.	Thank you for referring us to the study conducted by Lim et al (Anesthesiology, 2018 Feb;128(2):328-337) which we read with interest. We would stress the differences in the chosen methodologies of the two papers. Lim et al conducted a model based evaluation using data from published research or otherwise available in the public domain. In contrast SALVO was a rigorously conducted randomised controlled trial in clinical practice and we have identified the most pragmatic application of cell salvage and applied cost-effectiveness analysis. The Lim et al paper was not available at the time of submission for this paper. We now make reference to it in the discussion section (line 330) “A recent study by Lim et al found the use of cell salvage for cases at high risk for obstetric haemorrhage to be economically reasonable while routine cell salvage use for all caesarean deliveries was not. This study adopted a societal perspective and used data from published research to populate the model“

5	Page 13 Line 274-276. I am not sure this statement is substantiated by the data, especially because only the healthcare perspective was assumed in this particular study. Please, edit or remove.	Thank you for this comment. We have now revised the text on line 283 to read “Based on the results of this trial, in a UK setting the results suggest that cell salvage is not likely to be considered a cost-effective alternative to donor blood...”
6	The word “cesarean” is form the Greek “to cut” and thus the word “section” that follows is redundant as it translates to “cut cut.” Consider following all “cesarean” by the word “delivery” instead, throughout the text.	Thank you for this suggestion. The term ‘caesarean section’ is used in NHS literature and has been adopted in this study for that reason.
7	Methods. (Abstract and main text). It is easier for a reader if “Analysis 1, 2, and 3” can be termed something else that ties immediately to their group. E.g. “Analysis IIT” for intent to treat, “Analysis PP” per protocol, and “Analysis ECD” for emergency cesarean delivery. Please, consider.	Thank you for this helpful suggestion. We have revised the terms so they now read ‘Analysis ITT’, ‘Analysis PP’, and ‘Analysis ECD’
8	Methods. Some justification for why the perspective of health care provider, rather than societal perspective, was taken is desirable. Recommendations from the Panel on Cost-Effectiveness in Health and Medicine suggest that a more useful perspective from the standpoint of informing best practice is a societal perspective, rather than a healthcare-level perspective.	See comment 2 above
9	“Patient and public involvement” section. It is unclear why this section is here as the content is not germane to the economic analysis provided in the article. Consider removing, please.	This section is required for submission to BMJ open

Reviewer: 2

Reviewer Name: Steven M. Frank MD

Institution and Country: Johns Hopkins Medical Institutions, United States Please state any

competing interests: Advisory Boards for two companies that manufacture cell salvage equipment: Medtronic and Haemonetics

1	This randomized clinical trial reports the cost effectiveness of employing cell salvage for caesarean-section patients compared to the routine practice of giving allogeneic banked blood transfusions. Overall the study is large and well conducted, however some improvements in the paper listed below would be helpful to justify and support the main findings - that cell salvage was only marginally more effective in terms of avoiding allogeneic transfusion with uncertain cost effectiveness.	Thank you
2	3,028 women were enrolled with a transfusion rate of about 3-5%, depending on the group assignment. Of those assigned to the intervention group (cell salvage), 50.8% of those had salvaged blood returned with a mean volume of 260 mL. The problem is that these numbers are not apparent in the current paper under review but I obtained them from the authors' previous publication of the same study back in December, 2017 (Khan KS, Moore PAS, Wilson MJ, et al. Cell salvage and donor blood transfusion during cesarean section: A pragmatic, multicentre randomised controlled trial (SALVO) PLoS Med 2017;14:e1002471). The paper currently under review is the same exact study only it represents the economic analysis. One concern is that in the original paper on page 12 and in the S1 appendix (pages 2, 4, 6, 19, and 20) the authors already published much of the same economic data presented in the current paper. It would be nice to point out how these findings differ and warrant what could be considered re-publication.	It was not the intention of the authors to provide an economic analysis in the previous publication of the same study (Khan KS, Moore PAS, Wilson MJ, et al. Cell salvage and donor blood transfusion during cesarean section: A pragmatic, multicentre randomised controlled trial (SALVO) PLoS Med 2017;14:e1002471) but because there is a very short reference to the results of the intention to treat economic analysis the journal asked us to back this up with supplementary material. The methods and tables for this analysis were included as supplementary material with no interpretation of the results and with no further information about the additional two analyses (per protocol and emergency only) that were conducted in the full economic evaluation. What is included in the supplementary material does not adhere to the recommended standard reporting guidelines (CHEERS) and is therefore devoid of context, interpretation and implications. The current submission adheres to CHEERS guidelines and is a full analysis of the

		economic evaluation which is not found elsewhere Given the very helpful responses of the reviewers we can see how even when the full economic analysis is presented and the CHEERS guidelines are followed there can still be some questions about the analysis. We would therefore stress the value of the full economic analyses being published separately from the previous publication.
3	Although the findings are included in the previous publication, it would be helpful for the reader to see in the current paper the percentage of patients requiring transfusion of banked blood and the mean number of units transfused per patient in the two groups. I think table 1 may contain the percentage of patients transfused but it is difficult to figure out from the data in the table. I know the current paper is all about cost data, but some blood data also need to be included for clarity.	Thank you for this comment. As the reviewer noted, the information about the number of patients requiring banked blood is provided in Table 1 and the mean number of units transfused per patient is provided in Table 2. We have also provided a reference for the full details of the randomised controlled trial (line 117) should readers require more detailed information.
4	The inclusion criteria should be better described. For example, at LM 120, what were the identifiable risks of haemorrhage and the definition of an emergency caesarean section?	Thank you for this comment. The text on line 119 has now been revised to read: “The sample comprised women who were admitted to a participating labour ward and who fulfilled all of the following inclusion criteria:  • Aged ≥ 16 years • Able to provide informed consent • Undergoing delivery by caesarean section with an identifiable increased risk of haemorrhage, defined as all emergency caesarean sections, where maternal or fetal compromise is suspected, and elective caesarean

		section for all indications other than maternal request or breech presentation”
5	In the discussion, the authors may want to reference the following publication on the economic value of cell salvage in comparison to their findings (Waters JH, Dyga RM, Waters JF, Yazer MH. The volume of returned red blood cells in a large blood salvage program: where does it all go? Transfusion 2011;51:2126-32). This paper suggests that if one unit of salvaged blood is reinfused then the salvage procedure is cost effective, and the current study under review suggests this is not the case. Perhaps the difference is the high cost estimate in the current study of £1,244 extra cost per case where cell salvage is employed (stated on LM 204), a number which is not readily apparent from the data shown in Tables 2, 3, and 4. Does this number represent the incremental additional cost of employing cell salvage or just the average cost per case where it is used? The sentence starting on LM 203 doesn't make this clear. Also, does this relatively high cost include having a cell saver technician in the OR the entire case, or just for the initial setup and for processing the blood? This is really important because one cell saver tech can staff multiple ORs as they do not need to be in the OR the entire case. In fact, this variable could make or break the cost effectiveness of cell salvage and should be considered to be important for the current study. This detail should be added to the methods and discussion.	Thank you for referring us to the study conducted by Waters et al (Transfusion 2011;51:2126-32) which we read with interest. We suggest great caution in comparing that paper with the results here which is a full cost-effectiveness analysis. Whilst some comment on the cost of cell salvage is included in the Waters paper, the paper does not include a full economic analysis and references to cost saving are not based on a full economic evaluation. We appreciate your comment on the clarity of the cost per case. The text on line 209 has now been revised to read “The results of the ITT analysis suggest that routine cell salvage is more costly than standard care with the mean difference in total costs per patient estimated at £83.” Regarding the reviewers comment on the cost including a cell saver technician in the OR the entire case, the analysis includes the cost of additional staff called into theatre solely for the purposes of cell salvage and the amount of time they spent in theatre was recorded in the SALVO trial. This has now been clarified in Table 2. In sensitivity analysis (ii) the cost of additional staff was removed which marginally reduced the ICER.

	On LM 233 the sentence describes 202 centers in the trial but there were only 26 centers so this may be a typo.	Thank you for highlighting this error. The text on line 240 has now been revised to read “In the trial, 202 cases used a cell salvage machine...”
6	In this same sentence, a cell salvage machine is described that does not require consumables, and I think this sentence is misleading. All salvage machines require consumables that come in two packages. About 1/3 of the costs go towards the package that always needs to be opened and utilized which is the blood collection reservoir and the heparin solution to anticoagulated the shed blood. The other 2/3 of the costs goes towards the blood processing kit, containing the centrifuge bowl that the cell salvage machine uses for washing the shed blood. This package only need be opened if there is sufficient blood shed during the case to warrant processing. When a case is done under the “collect only” protocol, often only the first package need be opened initially, then in specific higher blood loss cases the second package can be opened. I think is what the authors mean to say when they describe a machine that doesn’t require disposables, which should be clarified. This collect only strategy is actually one that can substantially increase the cost effectiveness of using cell salvage and this should be recognized by the authors and perhaps more strongly emphasized.	Thank you for this comment. Yes, this statement relates to the fact that some machines were set up for collection only and others for collection and processing. The text on line 240 has now been revised to read “In the trial, 202 cases used a cell salvage machine that required consumables for collection only, even where the blood was not processed”
7	On LM 258 the authors describe variation in the estimated cost of blood and the potential cost increase to £1,500 for 3 units. It is unclear in the text, but what I think what the	Thank you for this comment but the reviewer is mistaken. We have assumed a cost of £1,500 and conducted threshold analysis to show the point at which routine cell salvage

	authors describe here is the full cost of an allogeneic blood unit including the overhead costs of managing that unit from donor to recipient - which is about 4 times higher than the acquisition cost of blood. The authors should reference this paper to support these overhead costs (Shander A, Hofmann A, Ozawa S, et al. Activity-based costs of blood transfusions in surgical patients at four hospitals. Transfusion 2010;50:753-65).	would be considered cost-effective. This is a widely accepted method in health economics. The text on line 265 has now been revised to read "Raising the cost of a three unit transfusion of RBC to a hypothetical cost of £1,500..." to clarify this for the reader.
8	On pages 30, 31, and 32, these three figures could all be condensed to one figure using 3 different colored lines to show the relative comparison between the three analyses.	Thank you for this useful suggestion. We have now condensed the three figures into one.

Reviewer: 3

Reviewer Name: Cyril Jacquot

Institution and Country: Children's National Health System, Washington, DC, United States Please state any competing interests: None declared

1	Did all women in the cell salvage arm receive reinfused cells or was there a minimum volume cutoff? For example, in a study by Milne et al. (Obstet Gynecol. 2015 Apr;125(4):919-23), only 21% of women had sufficient amount of intraoperative blood shed to proceed with reinfusion. In your study of patients with higher risk of bleeding, did the rate reach 100%?	Blood was returned if a sufficient volume was collected and processed. In the cell salvage arm 88.9% did not produce a sufficient volume of blood as referenced in the previous publication of this trial (Khan KS, Moore PAS, Wilson MJ, et al. Cell salvage and donor blood transfusion during cesarean section: A pragmatic, multicentre randomised controlled trial (SALVO) PLoS Med 2017;14:e1002471).
2	Page 7, line 120: Can you list the identifiable risk factors for haemorrhage used? Which ones had particularly good predictive values?	Thank you for this comment. The text on line 118 has now been revised to read: "The sample comprised women who were admitted to a participating labour ward and who fulfilled all of the following inclusion criteria:  • Aged ≥ 16 years • Able to provide informed consent

		 • Undergoing delivery by caesarean section with an identifiable increased risk of haemorrhage, defined as all emergency caesarean sections, where maternal or fetal compromise is suspected, and elective caesarean section for all indications other than maternal request or breech presentation”
3	Page 10, lines 195-196: In the UK, must blood consenting include discussion of alternatives to allogeneic blood (e.g. autologous blood)? Is that a factor in patients perceiving cells salvage as "reassuring, safe and preferable to donor blood transfusion"? A similar scenario I can think of in the United States is the Paul Gann Blood Safety Act in California: "a physician [...] shall inform [...] the patient of the positive and negative aspects of receiving autologous blood and directed and nondirected homologous blood from volunteers. For purposes of this section, the term "autologous blood" includes, but is not limited to, predonation, intraoperative autologous transfusion, plasmapheresis, and hemodilution."	In the UK, there is a duty for a clinician to provide a patient with accurate, up to date information about the proposed medical or surgical procedure. Informed consent for transfusion is required and the Test of Materiality applies: "The doctor is... under a duty to take reasonable care to ensure that the patient is aware of any material risks involved in any recommended treatment, and of any reasonable alternative or variant treatments. The test of materiality is whether, in the circumstances of the particular case, a reasonable person in the patient's position would be likely to attach significance to the risk, or the doctor is or should reasonably be aware that the particular patient would be likely to attach significance to it". Of course in practice you have the problem of labour pains, general anaesthesia and information overload to contend with, but these don't affect the underlying principles.
4	Page 11, lines 233-234 and page 13, lines 279-283: Cell salvage machines that "did not require consumables" and that "required different consumables" are described. I see the same effect on the ICER of £1,022 per donor blood	Thank you for this comment. This statement relates to the fact that some machines were set up for collection only and others for collection and processing. The text on line 240 has now been revised to read "In the trial, 202 cases used a cell salvage machine that

	transfusion avoided. Do these sections refer to the same alternative cell salvage machine? If necessary, please clarify the statements about the consumables.	required consumables for collection only where the blood was not processed”
5	Page 15, lines 309-310: I see the platelet/plasma use was not included. I was just curious if you would expect use to differ between patients receiving cell salvage versus donor blood transfusion. Might one group be at risk of more coagulopathy based on the red blood cells transfused?	The use of platelets and other blood products has not been included in the study. However, the results of the sensitivity and threshold analyses demonstrated that including these costs would not have impacted on the cost-effectiveness results. Platelets and clotting factors are removed from both donor and cell salvaged blood, so as long as the same volume is transfused, there would be no differential effect on the likelihood or severity of coagulopathy.

VERSION 2 – REVIEW

REVIEWER	Grace Lim, MD MS University of Pittsburgh, Magee-Womens Hospital of UPMC, USA
REVIEW RETURNED	08-Aug-2018

GENERAL COMMENTS	The revised manuscript is greatly improved. There are 3 points of major concern that remain to be addressed. Major Notes. P14L286-289. This statement is not substantiated by the results of the study. The ICER of 8110 per donor blood transfusion avoided is WELL below the willingness to pay threshold of 50,000, a threshold that was specified by the authors. Thus, the overall finding based on this ICER is that cell salvage IS IN FACT cost-effective when using this WTP threshold. HOWEVER, the sensitivity analysis suggests that the probability of IOCS in routine setting being cost-effective at this WTP threshold is 62%. This is not very robust. Thus, the confidence of the finding that it is cost-effective is not very high. According to the authors, the uncertainties are driven by lack of information on long-term outcomes. The statement currently talks about donor blood being freely available and supply is not restricted – these points are not germane to the principal findings of the study/analysis carried out, nor are they substantiated by the data provided in the study. It should be omitted. THEREFORE, please revise this statement to strictly reflect the findings of the study based on the analysis carried out, and include a statement about the lack of robustness of the findings based on the PSA of 62% Any additional conjectures about application to high resource vs. low resource centers should not be in the Principal Findings section.
--

	P17L357. Revise to state that the driver for uncertainty relates to the LONG TERM cost implications of adopting the routine use of cell salvage SPECIFICALLY with respect to QOL, etc. The figures for the cost-effectiveness acceptability curves appear unusual, and may represent an error in the use of the software to generating the curves. Please double check. An example of an appropriate curve is provided. The reviewer provided an example of an appropriate curve. Please contact the publisher for full details.
--	---

REVIEWER	Steven M. Frank MD Johns Hopkins University USA Advisory Board – Haemonetics
REVIEW RETURNED	08-Aug-2018

GENERAL COMMENTS	I would delete the sentence on LM 282-283 in the beginning of the discussion since it is redundant with the previous sentence. Otherwise the revision is well done.
---

VERSION 2 – AUTHOR RESPONSE

Reviewer: 1 Reviewer Name: Grace Lim MD MS Institution and Country: University of Pittsburgh, Magee-Womens Hospital, USA Please state any competing interests: None		
1	P14L286-289. This statement is not substantiated by the results of the study. The ICER of 8110 per donor blood transfusion avoided is WELL below the willingness to pay threshold of 50,000, a threshold that was specified by the authors. Thus, the overall finding based on this ICER is that cell salvage IS IN FACT cost-effective when using this WTP threshold. HOWEVER, the sensitivity analysis suggests that the probability of IOCS in routine setting being cost-effective at this WTP threshold is 62%. This is not very robust. Thus, the confidence of the finding that it is cost-effective is not very high. According to the authors, the uncertainties are driven by lack of information on long-term outcomes. The statement currently talks about donor blood being freely available and supply is not restricted – these points are not germane to the principal findings of the study/analysis carried out, nor are	Thank you for this useful comment. We agree with the reviewer that this point does not belong in the principal findings section and have opted to remove the statement on P14L286-289. The WTP threshold of £50,000 is an arbitrary threshold chosen to aid interpretation of the cost-effectiveness results and the CEACs. We have included the words “for example” on P11L226 to clarify this for the reader.

	they substantiated by the data provided in the study. It should be omitted. THEREFORE, please revise this statement to strictly reflect the findings of the study based on the analysis carried out, and include a statement about the lack of robustness of the findings based on the PSA of 62% Any additional conjectures about application to high resource vs. low resource centers should not be in the Principal Findings section.	
2	P17L357. Revise to state that the driver for uncertainty relates to the LONG TERM cost implications of adopting the routine use of cell salvage SPECIFICLLY with respect to QOL, etc.	We agree that long term should be included in the statement and thank the reviewer for this suggestion. This is now reflected on P17L358. The lack of QoL data is discussed on Pg15L320-322 as a limitation to this study along with the lack of information relating to the clinical status of the infant and that the use of other blood products has not been included in the study. We would therefore be hesitant to include the statement specifically with respect to QoL on P17L357. Instead, we have revised the statement on P17L359 to now read “Further studies should explore the long-term health, economic and quality of life impacts associated with both transfusion strategies. “
3	The figures for the cost-effectiveness acceptability curves appear unusual, and may represent an error in the use of the software to generating the curves. Please double check. An example of an appropriate curve is provided.	The figures used for the CEACs are the results of the three analyses conducted (1) Intention to treat analysis (2) Per protocol analysis (3) Emergency only analysis and the three curves show the reader how the results of each analysis compares to the other. This is different from the example provided by the reviewer where the results of an

		analysis are shown in comparison to usual care.
--	--	---

Reviewer: 2

Reviewer Name: Steven M. Frank MD

Institution and Country: Johns Hopkins Medical Institutions, United States Please state any competing interests: Advisory Boards for two companies that manufacture cell salvage equipment: Medtronic and Haemonetics

1	I would delete the sentence on LM 282-283 in the beginning of the discussion since it is redundant with the previous sentence. Otherwise the revision is well done.	Thank you. We have deleted this sentence.
---	---	---

VERSION 3 – REVIEW

REVIEWER	Grace Lim MD MS University of Pittsburgh, Magee-Womens Hospital of UPMC, Magee-Womens Research Institute, USA
REVIEW RETURNED	03-Dec-2018

GENERAL COMMENTS	The authors have performed an excellent revision of the manuscript.
---